# SoDaH: the SOils DAta Harmonization database, an open-source synthesis of soil data from research networks, version 1.0.

William R. Wieder[1], Derek Pierson[2], Stevan Earl[3], Kate Lajtha[2], Sara Baer[4], Ford Ballantyne[5], Asmeret Asefaw Berhe[6], Sharon A. Billings[7], Laurel M. Brigham[8], Stephany S. Chacon[2,9], Jennifer Fraterrigo[10], Serita D. Frey[11], Katerina Georgiou[12], Marie-Anne de Graaff[13], A. Stuart Grandy[11], Melannie D. Hartman[14], Sarah E. Hobbie[15], Chris Johnson[16], Jason Kaye[17], Emily Kyker-Snowman[11], Marcy E. Litvak[18], Michelle C. Mack[19], Avni Malhotra[20], Jessica A. M. Moore[21], Knute Nadelhoffer[22], Craig Rasmussen[23], Whendee L. Silver[24], Benjamin N. Sulman[25], Xanthe Walker[19], Samantha Weintraub[26]

[1]Institute of Arctic and Alpine Research, University of Colorado Boulder and the Climate Boulder, CO 80309, USA and Global Dynamics Laboratory, National Center for Atmospheric Research, Boulder, CO 80307, USA.

[2]Department of Crop and Soil Sciences, Oregon State University, Corvallis OR, USA and Idaho State University, Pocatello, ID, USA

[3]Global Institute of Sustainability, Arizona State University, Tempe, AZ, USA

[4] Department of Ecology and Evolutionary Biology and Kansas Biological Survey, University of Kansas, Lawrence, KS, USA

[5]Odum School of Ecology, University of Georgia, USA

[6]Department of Life and Environmental Sciences; University of California, Merced; Merced, CA, USA

[7]Department of Ecology and Evolutionary Biology and Kansas Biological Survey, University of Kansas, Lawrence, KS, USA

[8]Department of Ecology and Evolutionary Biology and Institute of Arctic and Alpine Research, University of Colorado, Boulder, CO, USA

[9]Climate and Ecosystem Sciences, Lawrence Berkeley National Laboratory, Berkeley, CA, USA

[10]Department of Natural Resources and Environmental Sciences, University of Illinois, Urbana, IL, USA

[11]Department of Natural Resources and the Environment, University of New Hampshire, Durham, NH, USA

[12]Department of Earth System Science, Stanford University, Stanford, CA, USA and Lawrence Livermore National Laboratory, Livermore, CA USA

[13]Department of Biological Sciences, Boise State University, Boise, ID, USA

[14]Climate and Global Dynamics Laboratory, National Center for Atmospheric Research, Boulder CO, and Natural Resource Ecology Laboratory, Colorado State University, Fort Collins CO, USA

[15]Department of Ecology, Evolution and Behavior, University of Minnesota, St. Paul, MN, USA

[16]Department of Civil and Environmental Engineering, Syracuse University, Syracuse, NY, USA

[17]Department of Ecosystem Science and Management, The Pennsylvania State University, University Park, PA, USA

[18]Department of Biology, University of New Mexico, Albuquerque, NM, USA

[19]Center for Ecosystem Science and Society and Department of Biological Sciences, Northern Arizona University, Flagstaff, AZ USA

[20]Department of Earth System Science, Stanford University, Stanford, CA, USA

[21]Bioscience Division, Oak Ridge National Laboratory, Oak Ridge, TN, USA

[22]Department of Ecology and Evolutionary Biology, University of Michigan, Ann Arbor, MI, USA

[23]Department of Environmental Science, The University of Arizona, Tucson AZ, USA

[24]Department of Environmental Science, Policy, and Management, University of California, Berkeley, CA, USA

[25]Climate Change Science Institute and Environmental Sciences Division, Oak Ridge National Laboratory, Oak Ridge, TN, USA

[26]National Ecological Observatory Network, Battelle, Boulder, CO, USA

*Correspondence to:* William R Wieder (wwieder@ucar.edu)

**Abstract.** Data collected from research networks present opportunities to test theories and develop models about factors

responsible for the long-term persistence and vulnerability of soil organic matter (SOM). Synthesizing datasets collected by different research networks presents opportunities to expand the ecological gradients and scientific breadth of information available for inquiry. Synthesizing these data are challenging, especially considering the legacy of soils data that has already been collected and an expansion of new network science initiatives. To facilitate this effort, here we present the SOils DAta Harmonization database (SoDaH; https://lter.github.io/som-website, last accessed Dec. 22, 2020), a flexible database designed

to harmonize diverse SOM datasets from multiple research networks. SoDaH is built on several network science efforts in the United States, but the tools built for SoDaH aim to provide an open-access resource to facilitate synthesis of soil carbon data. Moreover, SoDaH allows for individual locations to contribute results from experimental manipulations, repeated measurements from long-term studies, and local- to regional-scale gradients across ecosystems or landscapes. Finally, we also provide data visualization and analysis tools that can be used to query and analyze the aggregated database. The SoDaH v1.0

dataset is archived and available at https://doi.org/10.6073/pasta/9733f6b6d2ffd12bf126dc36a763e0b4 (Wieder et al., 2020).

# 1 Introduction

Soil organic matter (SOM) contains two- to three-times the amount of carbon (C) as the atmosphere and terrestrial vegetation combined, yet adequately describing SOM dynamics in numerical models remains a challenge (Jackson et al. 2017). Recent biogeochemical research has attempted to understand how climate, biota, soil chemistry, and mineralogy interact to determine

SOM stabilization and persistence (Schmidt et al. 2011; Lehmann & Kleber 2015). Emerging theories also highlight how interactions among these factors affect the production and apparent stabilization of microbial residues (Grandy & Neff 2008;

Cotrufo et al. 2013; Kallenbach et al. 2016). Notably, these new studies emphasize the importance of soil mineralogy and physical structure in limiting microbial access to otherwise decomposable substrates (Dungait et al. 2012; Miltner et al. 2012; Schimel and Schaeffer 2012; Sulman et al. 2014).

Datasets that span environmental and edaphic gradients are critical for constraining soil C estimates and developing and testing theoretical and numerical models that are based on these ideas (Wieder and Allison et al. 2015; Luo et al 2016; Harden et al. 2018; Sulman et al. 2018; Malhotra et al. 2019). Data synthesized across scientific networks, notably those with long-term observations and manipulations, are especially useful for establishing general patterns across broad environmental gradients. These insights, and the primary data are valuable for model development. For example, efforts to synthesize and

archive results from the Long-Term Intersite Decomposition Experiment Team (LIDET; Gholz et al. 2000; Parton & Silver et al. 2007; Adair et al. 2008; Harmon 2013) provide a valuable benchmark for parameterizing and evaluating models with litter decomposition data (Bonan et al. 2013; Wieder and Grandy et al. 2015; Kyker-Snowman et al. 2019). Elsewhere, Zhang et al. (2020) used data from three research networks in Europe, China, and Australia to parameterize and evaluate two soil carbon models. Providing similar data syntheses with information on soil carbon and associated covariates (e.g., climate, productivity,

and soil physical and chemical properties) in public databases is critical to advancing understanding soil biogeochemistry.

        Coordinated research activities and the expansion of research network infrastructure are broadening the scope and breadth of information measured across sites in ways that can advance SOM science (Hinckley et al. 2016; Baatz et al. 2018; Richter et al. 2018; Weintraub et al. 2019, Lajtha et al. 2018). With a 40-year investment in continuous or multi-year measurements and a rich legacy of manipulative experiments, the Long-Term Ecological Research (LTER) Network provides

a publicly available data archive through the Environmental Data Initiative (EDI; https://portal.edirepository.org/nis/home.jsp). The LTER network has an advantage of hosting diverse research experiments, but because each site in the network has different research foci data are not collected or reported in a consistent manner (Billings et al. 2020, but see Zak et al. 1994; Frank et al. 2012). By contrast, new investments in networks like the National Ecological Observatory Network (NEON) provide a top-down, standardized framework for data collection across sites.

Synthesizing data from across LTER, NEON and other research networks present unique opportunities to deepen our general understanding of soil biogeochemistry.

        Here, we present a flexible database designed to harmonize diverse SOM datasets from across research networks. We aim to provide an open-access resource to facilitate the synthesis of soil C data. This data resource can expand to accommodate legacy datasets as they are identified and incorporate new data products as they become available. This data infrastructure is

critical to advance understanding in SOM dynamics at a time when the theoretical foundations and numerical representations of soil biogeochemical processes are rapidly evolving.

## 2 The SoDaH database

Our team created the SOils DAta Harmonization (SoDaH) database to bring together soil C data from diverse research networks into a harmonized dataset that can be used for synthesis activities and model development. The research network sources for SoDaH span different biomes and climates, encompass multiple ecosystem types, and have collected data across a range of spatial, temporal, and depth gradients. The rich data sets assembled in SoDaH consist of observations from monitoring efforts and long-term ecological experiments. The SoDaH database also incorporates related environmental covariate data pertaining to climate, vegetation, soil chemistry, and soil physical properties. The data are harmonized and aggregated using open-source code that enables a scripted, repeatable approach for soil data synthesis. Finally, to accompany SoDaH, we provide data visualization and analysis tools that can be used to query and analyze the aggregated database.

### 2.1 Database Sources and Structure

Research networks provide a powerful observational platform for enhancing our understanding of ecosystems. For example, in the United States, three research networks funded by the National Science Foundation collect soils data that deepen understanding and improve the representation of soil biogeochemical processes in models. These include the LTER network (https://lternet.edu/), Critical Zone Observatories and their successor sites (CZO; http://criticalzone.org/national/), and the National Ecological Observatory Network (NEON; https://www.neonscience.org/, NEON 2020). Other coordinated research activities that further expand data availability include community efforts like the Nutrient Network (NutNet; https://nutnet.org/) and Detritus Input and Removal Treatments (DIRT; https://dirtnet.wordpress.com/). We compiled soils data from these five research networks into the SoDaH database, version 1.0.

The unique perspectives and historical legacies of each network synergistically offer insights into understanding many aspects of SOM dynamics. For example, data from LTER, DIRT and NutNet sites are generally long-term datasets that focus on surface soil (< 30 cm) properties across gradients and response to experimental manipulations. Data from CZO sites tend to contribute information on soil geochemical properties and expand focus to include deeper (> 30 cm) soil horizons. Finally, NEON employs standardized data collection procedures that span continental-scale ecoclimatic gradients (Fig 1).

The SoDaH dataset focuses on soil organic carbon (SOC) concentration (% C), estimated SOC stocks (g C m$^{-2}$), and associated covariates that may be useful in explaining variation in SOC stocks within and among sites. To avoid confounding the interpretation of SOC measurements collected by different approaches (e.g. Walkley-Black and mass loss on ignition), we focused on synthesizing SOC measurements from soil samples that were acidified if needed to remove inorganic carbonates, then analyzed for total C using elemental analyzer. Beyond SOC, covariates collected in SoDaH include abiotic factors (e.g., climate [mean annual temperature and precipitation], soil depth, bulk density, particle size distribution, and mineralogy), vegetation characteristics (including vegetation type and above and belowground root productivity, biomass, and chemistry), and additional soil chemical properties (total nitrogen, phosphorus, pH, etc.).

Recognizing that the cyber landscape of soil databases is expanding (Malhotra et al. 2019), we wanted to structure SoDaH in a manner consistent with existing databases, perhaps most notably the International Soil Carbon Network (ISCN; Nave et al. 2016, Harden et al. 2017), which similarly focuses on SOC concentrations and stocks in bulk soils. The ISCN uses a hierarchical data structure that links metadata information with fields for location, profile and soil layer data. We maintained the ISCN's basic structure in SoDaH (Fig. 2), as it provides a logical means to structure relationships between different measurements (i.e., variables). A similar approach was also used in the International Soil Radiocarbon Database (ISRaD; Lawrence et al. 2020), which primarily focuses on synthesis of additional information about radiocarbon from bulk soils, soil fractions, and soil gases. Given this focus of ISRaD, the SoDaH database contains only sparse data on isotopes and SOM fractions. Since SoDaH and ISCN focus on SOC measurements and have a similar structure, we hope they may be used together in future studies.

The unique contribution from SoDaH, relative to other soil databases, is that SoDaH is built on several network science efforts in the United States, and presents a usable, extensible database for contributing and analyzing data. Moreover, SoDaH allows for individual locations to contribute results from experimental manipulations, repeated measurements from long-term studies, and local- to regional-scale gradients across ecosystems or landscapes. Data from these kinds of studies should be incorporated into existing database structures, like ISCN, but the additional metadata requested as part of SoDaH helps database users understand more information about how data were collected from individual studies. Thus, SoDaH allows for the harmonization of data spanning a greater range of spatial and temporal scales than other databases, and enables the incorporation of ecosystems responses to manipulations, which is not a possibility for other databases.

Given the focus on experimental manipulations, we requested additional categorical information on location and profile fields to clarify aspects of data collection and experimental design. This includes flags in the location field asking if datasets include measurements that are repeated over multiple time points, come from experimental manipulations, or represent gradient studies. We also asked dataset contributors to identify 'control' or unmanipulated sample identifiers when necessary. We accommodated various experimental designs and data hierarchies with fields to describe this information, such as whether plots are grouped into blocks or watersheds, and the organization of treatment levels, in the profile field of the database. For example, at one site, data may be collected from plots along an elevational transect; whereas, another dataset may include information from a nitrogen fertilization treatment that was conducted on experimental plots in a replicated block design. Maintaining these data hierarchies is important for database users to inform how best to aggregate data collected from diverse networks, individual study sites, and unique experimental designs.

The workflow for synthesizing is summarized in Figure 3 and in the following sections. Briefly, Primary data (Level-0) are identified by data providers and variables are mapped to standardized units and vocabulary using the metadata templates (section 2.2). These data are harmonized into Level-1 data with soil harmonization script that renames variables, conducts unit conversions, and performs quality control checks (section 2.3). Finally, Level-1 data are aggregated into the Level-2 dataset, which can be visualized with the SoDah R Shiny app and queried with data analysis tools (section 2.4).

## 2.2 Data Identification and Contributions

To begin populating the SoDaH database, we identified data contributors who were familiar with primary datasets available from individual study sites and research networks. These primary data may or may not be in a published state but, if not published, would be equivalent to data provided for publication. Many of the datasets in SoDaH were already published in public repositories like EDI, the repository for LTER data, or available through the NEON data portal. Users can find these primary data using the doi provided for individual dataset in the harmonized dataset. Other datasets that we wanted to include in SoDaH, however, had not been published or were difficult to find or identify (mainly data from CZO sites and the DIRT network, but also some LTER data). Publishing these primary data remains an active priority for our working group. Data providers who were familiar with the diversity of datasets that are available at a study site or a network provided expertise to link soil C datasets with appropriate ancillary data.

The SoDaH database was constructed by data contributions from individual sites or research networks who provided flat (.csv) files to a shared directory on Google Drive. The dataset (or datasets) from each site, study, or network were placed in their own subdirectory along with a metadata template that was used to map variable names in the primary (Level-0) data to the structure of SoDaH (Fig. 3). The metadata template was developed to facilitate data harmonization in a scripted, repeatable manner that maintained the integrity of the primary datasets (https://lter.github.io/som-website/database.html). To simplify the workflow for data contributors, the metadata template only includes a single tab each for location and profile data. Within these tabs, data contributors are able to add information on metadata (found on the 'location' tab) and layer or fraction data (found on the 'profile' tab; Fig. 2). Layer data includes information on soil chemical and physical properties that may be measured on bulk soils for defined soil horizons or depth increments. Fraction data would include similar measurements on defined fractions within individual soil layers (e.g., percent soil organic carbon on density fractionated soils). Note, SoDaH currently has sparse data from measured soil fractions, which have therefore been omitted from Fig. 2 for simplicity, but the database structure can include information on soil fractions.

This initial step of our data harmonization still requires manual effort from data providers, as they have to map the names of measured variables from primary data with the appropriate variable in SoDaH. Data contributors enter relevant metadata and site information that may not be included in the primary data sets. They provide additional information from controlled drop-down cells with information on units for each variable (e.g., %C, g C kg$^{-1}$ soil, mg C kg$^{-1}$ soil, etc.) or on methodologies used (e.g., soil P measured by Bray, Melich, etc.). In the harmonized dataset, we convert analyte names and units to a standard output and include methodological information (section 2.3). This approach accommodates a broad suite of soil and related variables (e.g., climate, vegetation characteristics, ecosystem productivity, etc.). In the future, we aim to further reduce data provider input requirements, but only if the community converges on standardized variable names and units of measure (*sensu* Billings et al. in press). Ultimately sophisticated metadata, such as controlled vocabularies and other, more expressive semantic technologies, may facilitate scripted harvesting of data from disparate networks and repositories (e.g., see review by Buck et al. 2019 for trends and examples in Marine Science).

The metadata template in SoDaH matches site-level information with the detailed measurements collected at each study site. Data on the location tab represents site characteristics for a single site or location (e.g., Prospect Hill Warming experiment at Harvard Forest). Accordingly, the harmonization script broadcasts data provided on the location tab (latitude, longitude, mean annual temperature, etc.) to every row of the harmonized dataset. Data on the profile tab includes profile information about experimental levels (e.g., plots within experimental blocks) and experimental treatments (e.g. +N

fertilization) that help clarify how the data were collected. Data on the profile tab should also correspond to columns of variables that are reported in the Level-0 data (e.g., soil organic C measured at different soil layers). Accordingly, the harmonization script copies each unique measurement from the profile tab into a column of data in the harmonized dataset. Data contributors, therefore, can move variables from the location to profile tabs when appropriate. For example, NutNet and NEON data were submitted to SoDaH with information from multiple sites on a single .csv file that provided information

about each site as unique columns of data.  We, therefore, moved site information (e.g., climate, latitude and longitude) onto the profile tab for these networks.  Similarly, gradient studies that report tabular data for individual soil profiles can move information on slope, aspect, vegetation communities or parent material (typically on the location tab) onto the profile tab of the metadata template.

        The harmonization script can harmonize multiple datasets from the same study location. For example, a dataset may

consist of multiple data files that each contain details about different aspects of the study (e.g., soil data in one file, aboveground productivity in another file); the harmonization script will harvest all variables identified in the metadata file from the suite of data files (as long as they are in the same Google directory as the metadata file). However, because SoDaH is a flat database values from these different data files will be stacked, meaning that information from different Level-0 datasets would be recorded in different rows of the aggregated Level-2 database (in the example above, soil properties and

productivity will be included, but in different rows). Additional aggregation steps, therefore, may be required to align data within sites. Users can find this information in the database column labeled *merge_align*, which is a logical that identifies if multiple data files can be merged.  Notes under columns *align_1* and *align_2* are intended to help communicate what common data fields can help with this alignment (e.g. experimental or treatment levels, *L1* and *tx_L1*, respectively). To help users understand the database column information, the complete database key is provided in the SoDaH online application

and gives users descriptions of the column contents.

## 2.3 Data Harmonization and Aggregation

We developed the *soilHarmonization* package in R (R Core Team 2020) to harmonize and aggregate the SoDaH database. The *soilHarmonization* package is publicly available (https://github.com/lter/soilHarmonization).  The package includes functions that harmonize Level-0 data into Level-1 data. Data contributors or database managers use the *data_harmonizaiton*

function tools to read and harmonize user-provided primary data that are mapped to a metadata template with controlled vocabulary and standard units (Fig. 3). Users point to the Google Drive directory where Level-0 data are located (primary

data and metadata template), and the *data_harmonizaiton* function generates a new flat file(s) in which the variable names and units are standardized in the output (Level-1 data). The harmonized dataset includes unique columns of data from those defined in the profile tab as well as columns of data with site-level information from the location tab. The package also

includes a suite of QC tools that confirm proper data type (e.g., strings are not interspersed with numeric values) and that numeric data, once converted to appropriate units, fall within an expected range. A summary of inputs, outputs, harmonization steps, and a QC report are detailed in an accompanying document (.pdf) for each harmonized dataset. These Level-1 data products are stored in the same Google Drive directory as the Level-0 data with resulting output identified with a modified filename. This allows data contributors and database managers to verify the QC report and ensure appropriate

data harmonization.

After generating Level-1 data from all Level-0 data, we combined harmonized data files into an aggregated dataset (.rds  or .csv  format; Fig. 3). This *dataHarvest* function is intended for use by database managers and is available on the LTER SOM GitHub page (https://github.com/lter/lterwg-som/tree/main/data-aggregation/, last accessed Dec. 22, 2020). This function aligns columns of Level-1 data into a single, Level-2, dataset. The resulting SoDaH database (version 1.0) we

describe here is a single, flat dataset that has columns corresponding to variables in the metadata template and rows for each measurement.

## 2.4 Data Visualization and Analysis

To facilitate user interaction with the SoDaH database, and to provide a simplified approach for data queries and analysis, we developed a web-based application using R Shiny (Chang et al. 2020). This SoDaH application is publicly accessible and

hosted by the National Center for Ecological Analysis and Synthesis (NCEAS) at https://cosima.nceas.ucsb.edu/lter-som (last accessed Dec 22, 2020; source code: https://github.com/lter/lterwg-som-shiny). With the SoDaH application, users can perform a number of tasks to aid data discovery, visualization and analysis. We provide a brief description of this resource that highlights key features of the R Shiny SoDaH application.

In the *Query* section of the application, the top portion of the page provides a variety of data filter options to assist

users with partitioning the database. Specifically, users may subset the database by any combination of research network, experiment type, and soil depth, while also specifying whether they wish to include or exclude experimental treatments or time-series data.  Below the filter options, the *Output* section of the page contains three separate features arranged into labeled application tabs. The *Plot* tab allows users to quickly create basic analysis plots (point, histogram, or boxplot) using both covariates (e.g., Fe concentration) and metadata (e.g., mean annual precipitation). In the *Map* tab, users may specify

which analyte in the database to display on a spatial map. Numeric values are symbolized using a color gradient and the interactive map functionality allows users to both adjust the map scale and select from numerous basemap options. Finally, the *Table* tab provides users with the ability to directly view, search and download the user-specified data subset as a flat file

(.csv). The plot, map and table features are all responsive to user specified changes in the data filters and will update in realtime.

The data table on the *Query* page of the SoDaH Shiny application is responsive to the filter options at the top of the *Query* page. When users click the "Download data" button next to the table, the downloaded .csv file will contain the same data shown in the application table at that time. Code examples for working with the database, including how to filter by specific column values, are provided in the GitHub repository (https://github.com/lter/lterwg-som/data-processing/Tarball_v2 scripts, last accessed Dec. 22, 2020).

In the *Data Summary* section of the SoDaH application, two feature tabs are provided to help users identify the data available for a specific site or analyte. The *By Analytes* tab allows users to view the number of analyte values that exist across all of the unique sites in the database. Users may specify up to four different analytes at a time to be included in the summary table output. The *By Site* tab allows users to view all of the analyte data available for a specific site. As the amount of data may be quite large for some sites, options are provided to narrow the summary output to include only profile,

location or character class data.

        The SoDaH application also includes a *Data Key* section, where users may view a full copy of the metadata template used for the SoDaH database construction, including descriptions of database fields and their associated metadata. The searchable key is split into two sections, location and profile, in the same manner as the metadata template used to describe primary data for the harmonization process. Field names in the provided key match exactly with analyte and

metadata options provided in the *Plot* and *Map* features in the *Query* section of the application. Finally, the application provides a *Comments* section where users may submit an inquiry about the database or the application.

        For users seeking to move beyond the functionality provided by the SoDaH application, R scripts are provided through the LTER SOM GitHub repository (https://github.com/lter/lterwg-som/tree/main/data-projects, last accessed Dec. 22, 2020) to facilitate and demonstrate scripting language to import, filter, summarize and map data from the SoDaH

database. This repository is intended to facilitate use of the SoDaH database, and the scripts used to generate figures in this paper are available in the repository. We encourage database users to draw from these existing resources and contribute new scripts they develop for scientific analysis of data in SoDaH.

        Additional data aggregation steps may be required to fully realize strengths of the SoDaH database. These could include, identifying suitable approaches to aggregate, and aligning data within sites. The aggregation steps currently

implemented in SoDaH may not be appropriate for particular research questions, especially those concerning spatial and temporal gradients. Therefore, users may need to align rows of data from the same profile or location, but were harvested from multiple data files, which results in data being stacked within the flat database. For example, a site may contribute data on soil chemical properties, soil physical properties, microbial stoichiometry and biomass, litterfall chemistry, and litterfall fluxes with each as an independent dataset. Moreover, these variables may be measured multiple times during a long-term

study, but not necessarily at the same time or at the same frequency. Finally, information from a single site may include a gradient study across a hillslope, chronosequence, or region that may influence how data users want to aggregate individual

measurements. The SoDaH metadata template prompts data providers to indicate if data from multiple files need to be aligned, and, if so, the grouping variable(s) that can be used to join this information (see section 2.2). The template also prompts data providers to indicate if datasets include time-series data or data from a gradient study. Users of SoDaH are encouraged to consider this information in their analyses.

## 3 Database description

### 3.1 Spatial and temporal distributions

The SoDaH database currently contains data from 215 locations and 186 unique study sites, with data contributed from DIRT, NutNet, LTER, NEON, and CZO networks. There are more locations than study sites in the database because some sites contributed datasets from multiple locations or experiments. The flat database contains 160 columns of variables and nearly 300,000 rows of information, but is relatively sparsely populated, with 13.9 million non-missing observations (roughly 30% of the database). Given the focus on NSF funded research networks and observatories, most of the measurements are taken from the United States, but NutNet and DIRT networks include a number of international study sites (Fig. 4).

Mean annual temperature from all locations was $10.1 \pm 7.1$ °C (mean $\pm 1\sigma$, n = 212) with a range of -12 to 27.2 °C. Mean annual precipitation from all locations was $904 \pm 638$ mm y-1 (n = 213), with a range of 105 to 4250 mm y-1. Land cover classifications include urban, cultivated, rangeland/grasslands, shrublands, and forests, but land cover is reported only for a subset (n = 87) of the study locations.

We briefly review characteristics of data contributed from the five networks represented in SoDaH (Fig. 5). The CZO generally has a focus on making one-time characterizations that extend deeper in soil and regolith profiles than other networks. Data from DIRT spans relatively few sites and only includes surface soil layers, but provides repeated measurements and their response to experimental manipulations. The LTER network provides data from comparatively few study sites, but LTER sites have longer measurement records than other networks in SoDaH given the network's 40-year history. Some data from LTER sites also include measurements to ~1m depth. By design, NEON provides data with broad geographic coverage and samples both surface and deeper soil horizons. The current temporal record from NEON sites is relatively short, but is expected to extend for the next 30 years. Finally, NutNet provides the greatest number and largest spatial distribution of sites, all from grassland ecosystems with sampling depths from 0 to 10 cm.

### 3.2 Experimental manipulations, gradients, and time series

SoDaH is unique in the landscape of soil databases because it includes data from both experimental manipulations (at 132 sites) and gradient studies and includes time series of soil data. Nutrient manipulations from NutNet make up the majority (109) of experimental manipulations. All experimental manipulations in SoDaH are summarized in Table 1 and include

manipulations from all fifteen LTER sites for which we have data, six DIRT sites and one CZO site. The database also includes gradient studies from 66 sites (with data from NEON, CZO and LTER networks), and time series data from 158 sites (with data from NutNet, NEON, LTER, and DIRT networks, Table 1).

## 330 3.3 Database use and analyses

Aggregating data in SoDaH presents challenges in how to most appropriately group multiple measurements taken from individual study locations that include diverse sampling protocols, unique experimental designs, and measurements from multiple soil depths. Moreover, particular locations may include manipulative experiments, gradient studies, and time series of repeated measurements. The appropriate aggregation of SoDaH requires users to become familiar with data structures of

the database to address particular scientific questions. For this reason, we see the RShiny web-app as an invaluable tool for querying the data available from SoDaH. As mentioned in section 2.4, future contributions of code to analyse the SoDaH database are encouraged. These contributions should be made to the LTER SOM GitHub repository, with a priority on developing additional utilities to align and aggregate datasets from individual sites and locations. Contributions will be reviewed by the SoDaH steering committee (currently Wieder, Pierson and Earl) and made publicly available. The committee

will continue oversight while new funding options and/or partnerships (e.g., ISCN) are explored.

## 3.4 Database contributions and database versioning

We built the SoDaH tools to help facilitate the harmonization of diverse soils datasets that focus on soil C. Towards that end,

we welcome contributions of new data from new sites that may be part of the research networks presented here, additional research networks (e.g. Ameriflux https://ameriflux.lbl.gov/, Drought-Net https://wp.natsci.colostate.edu/droughtnet/, Long-Term Agroecosystem Research https://ltar.ars.usda.gov, African soils database http://africasoils.net/services/data/, European LTERs https://www.lter-europe.net/, or others), as well as data from sites that are unaffiliated with a research network. The SoDaH website (https://lter.github.io/som-website/database.html, last accessed Dec. 22, 2020) contains more information on

how to contribute data. Briefly, data contributors need to place primary datasets and a completed copy of the SoDaH metadata template into a shared Google Drive folder and notify the SoDaH editor (soildataharmonization@gmail.com) that their data are ready for ingestion into SoDaH. These data contributions will also be reviewed by the SoDaH steering committee. We ask that new contributions of primary data that are harmonized into SoDaH be published with a unique DOI.

Updated releases of SoDaH will be made periodically after a threshold number of new contributions have been made

to the database, in light of any changes to the database structure, or if any errors are detected and corrected. Versions are tracked with a version number in the form of "major.minor." in addition to the date of publication. Each version of the dataset will receive a unique citation and DOI through the EDI data portal for users to reference.

## 4.0 Data availability and user guidelines

The SoDaH v1.0 database and some exemplary analyses are hosted in the EDI repository (Wieder et al., 2020; https://doi.org/10.6073/pasta/9733f6b6d2ffd12bf126dc36a763e0b4 accessed Dec. 22 2020). We encourage users of SoDaH data to cite both this publication and the dataset citation provided by the EDI data portal in their products.

**Author contribution:** WRW and KL received funding for the synthesis. WRW, SE, and DP designed the approach harmonized datasets, and published the synthesis. All other authors contributed data to the synthesis and provided input on this manuscript.

**Competing interests:** The authors declare that they have no conflict of interest.

### Acknowledgements

This paper stems from a synthesis group Advancing Soil Organic Matter Research: Synthesizing Multi-scale Observations supported through the Long-Term Ecological Research Network Office (LNO; NSF award numbers 1545288 and 1929393) and the National Center for Ecological Analysis and Synthesis, UCSB lead by KL and WRW. WRW was also supported by the Niwot Ridge LTER program (NSF DEB – 1637686), SE by the Central Arizona–Phoenix LTER program (NSF DEB – 1832016), DEB-1257032 to KL, and DEB-1440409 to the H. J. Andrews LTER program.

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

**Table 1. Summary of the networks and number of sites contributing data from experimental manipulations, gradient studies, and time series of repeated measurements. Gradient studies may include measurements along a hillslope catena (e.g., several CZO sites), across vegetation communities (typically LTER sites), or surveys intended to capture local- to regional- variability (especially NEON periodic soil sampling). Time series studies involve repeated measurements in the same sites over time (LTER and NEON) and they which may also include experimental manipulations (e.g., NutNet, DIRT, & LTER).**

| Experimental Manipulation | Networks (site) |
|---|---|
| Nutrient additions | NutNet (109) LTER (5) |
| Litter manipulations | DIRT (6) |
| Agricultural management | LTER (3) |
| Forest harvest | LTER (2) CZO (1) |
| Warming | LTER (2) |
| Fire | LTER (2) |
| Precipitation manipulation | LTER (2) CZO(1) |
| Elevated $CO_2$ | LTER (1) |
| Other (mostly related to management, disturbance, or land use history) | NutNet(109) LTER (10) CZO (1) |
| **Gradient Studies** | NEON (47) LTER (11) CZO (7) |
| **Time Series** | NutNet(109)^ NEON (35)§ LTER (10) DIRT (5) |

^ Repeated measurements for NutNet are for plant productivity, not soil measurements

§ Not all NEON sites have been sampled more than once per dataset

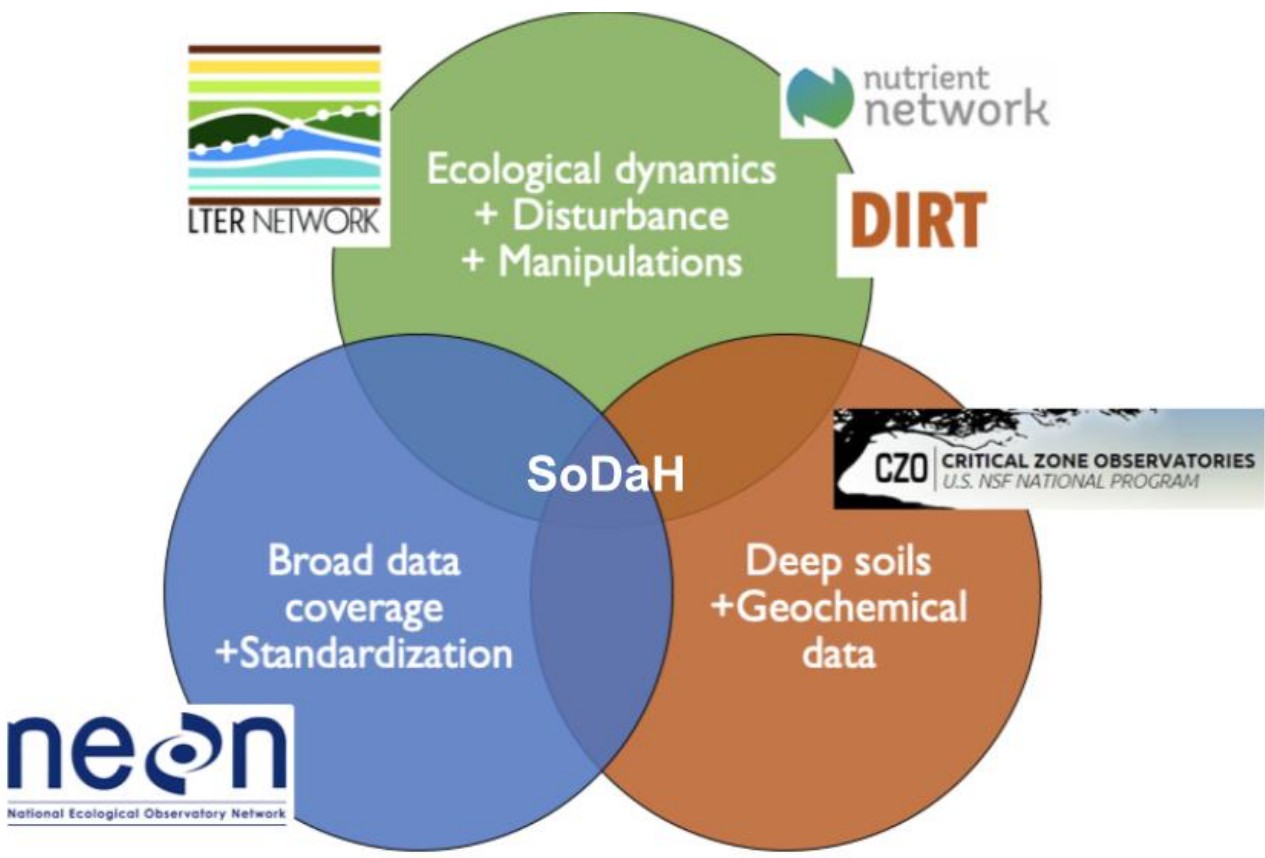

**Figure 1: Conceptual diagram that summarizes the strengths and research foci of different experimental networks contributing to SoDaH, modified from Weintraub et al. 2019.**

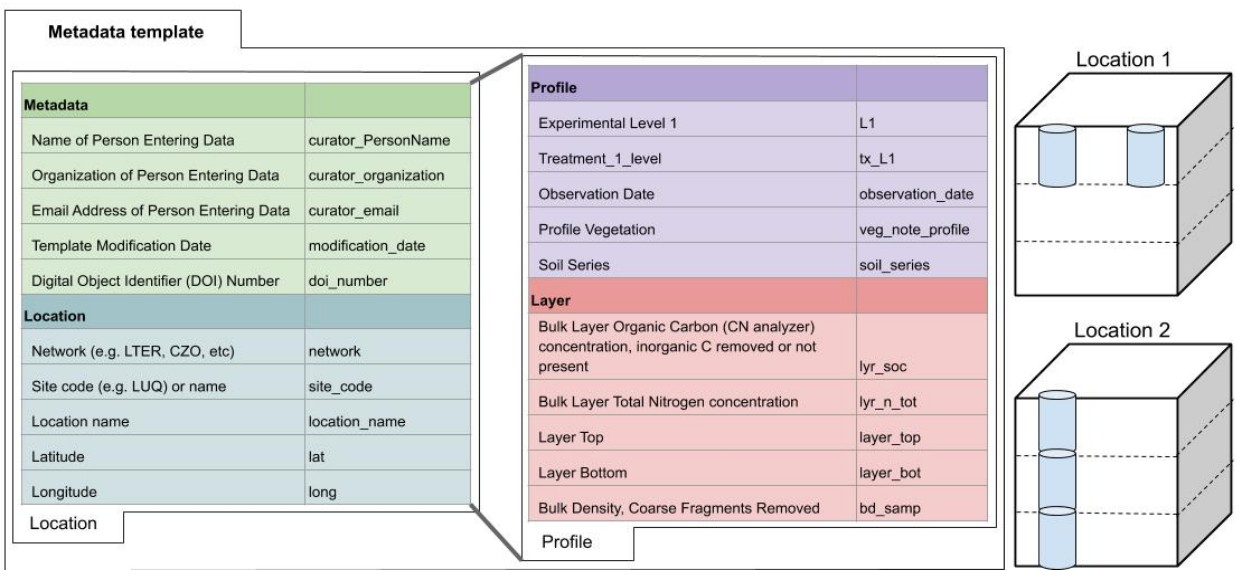

**Figure 2: Diagram showing hierarchical relationship between data fields in the Soils Data Harmonization (SoDaH) database, which includes metadata, location, profile and layer fields. Each data field lists a short description of some of the variables used along with the variable name used in the database. To facilitate data contributions these data fields were grouped into Location and Profile tabs on the metadata template used by data contributors. The right side of the figure illustrates data from two hypothetical locations (e.g., a LTER and CZO site, respectively) where Location 1 includes data from two profiles that each have**
**information from one layer. Location 2 provides data from one profile that has information from three layers. Any location may provide data from multiple profiles or layers. With data harmonization data for each profile and layer will inherit metadata and location data that are provided in the location tab.**

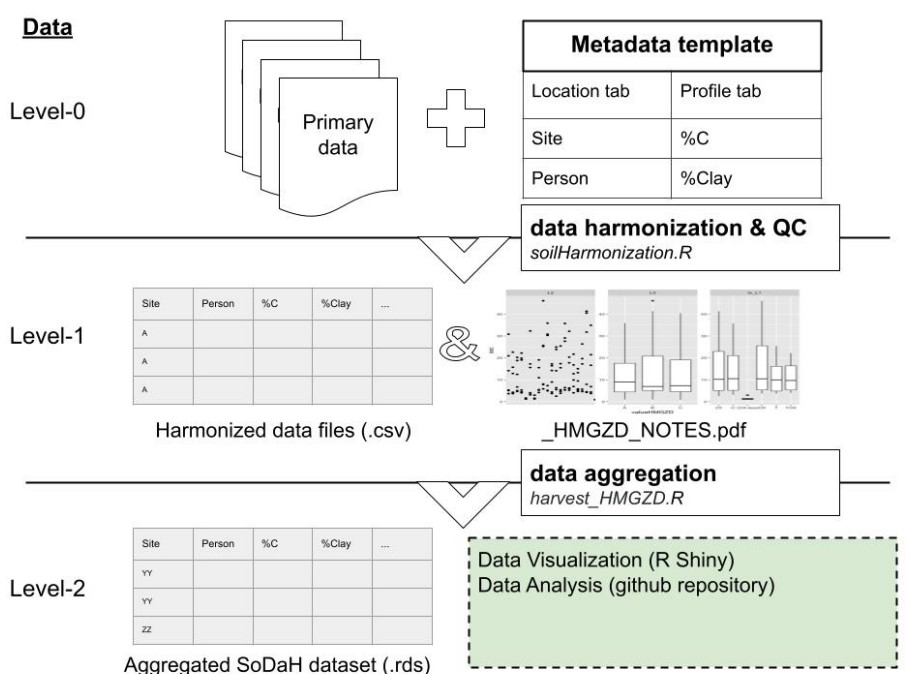

**Figure 3: Illustration of the SoDaH workflow and data levels. Primary data (Level-0) are identified by data providers and variables are mapped to standardized units and vocabulary using the metadata templates. These data are harmonized into Level-1 data with soil harmonization script that renames variables, conducts unit conversions, and performs quality control checks. Finally, Level-1 data are aggregated into the Level-2 dataset, which can be visualized with the SoDah R Shiny app and queried with data analysis tools.**


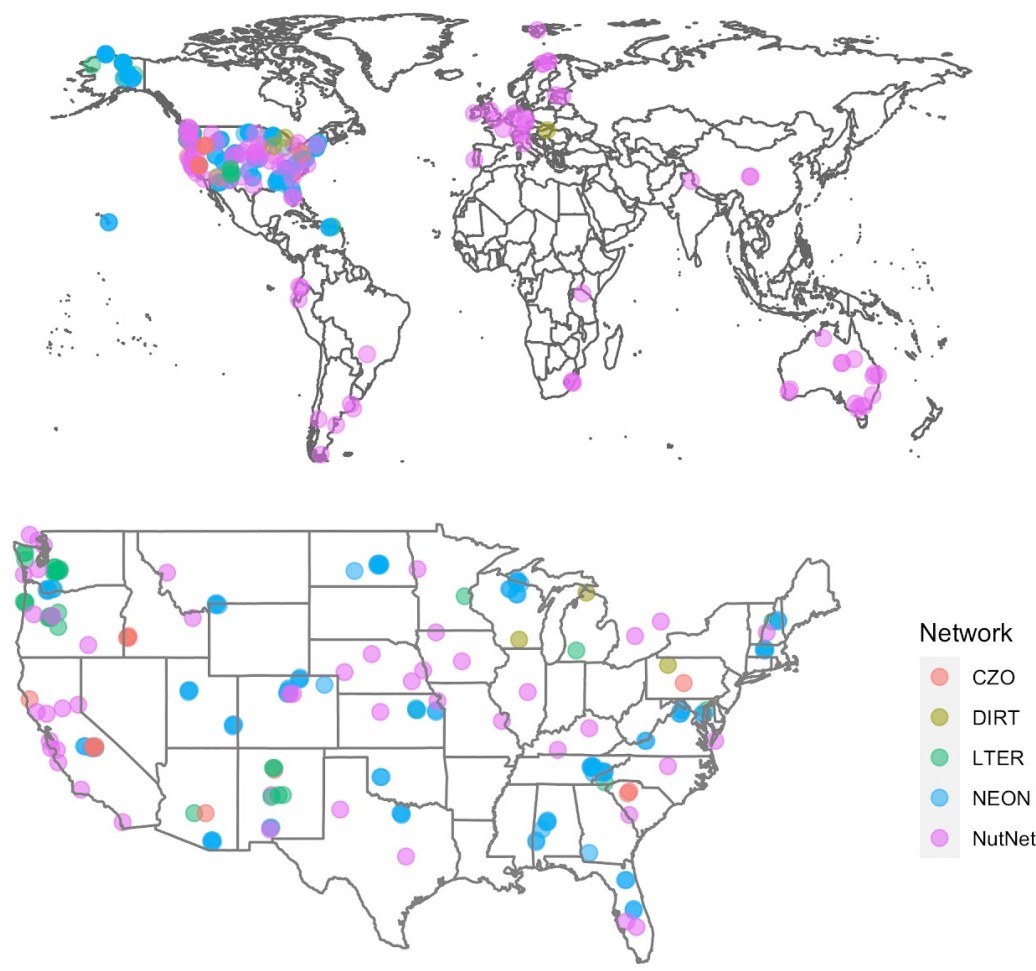

**Figure 4: Spatial distribution of study locations representing five research networks in SoDaH globally and in the contiguous USA.**


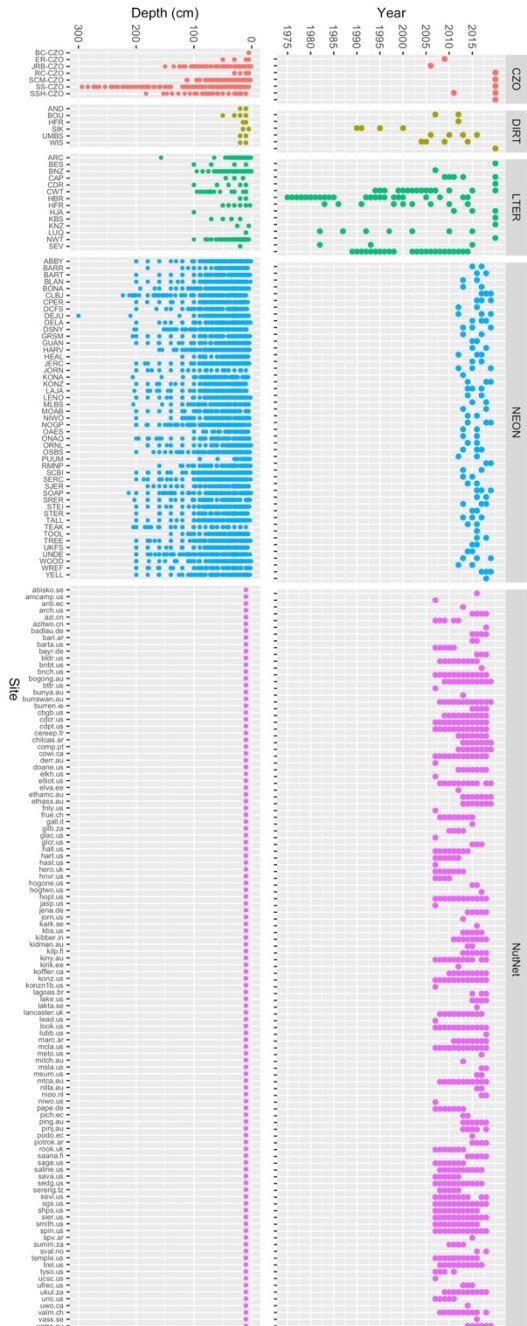

**Figure 5: Temporal coverage and depth of measurements taken from different study sites and grouped by research network. Our intent with this figure is to illustrate the number of sites in each network, the temporal length of their data record, and the depth to which soils are typically sampled.**
