# Peer review of "SoDaH: the SOils DATA Harmonization database, an open-source synthesis of soil data from research networks, version 1.0."

_Earth System Science Data, 2020_

## Referee Comment (RC1) · Jeffrey Beem Miller (Referee) · 10 Nov 2020

General comments

The authors present a new database (SoDaH) of soil observations synthesized from datasets curated by five well-known research networks: LTER, CZO, NEON, NutNet, and DIRT. Two key aspects of SoDaH make it unique among the new generation of soil databases: 1) flexibility in its approach to harmonizing data from diverse sources, and 2) the development of a powerful web-based tool for querying, visualizing, and extracting subsets of data. These two features of SoDaH make it a valuable addition to the growing pool of soil databases available to the soil science community.

The manuscript is generally well written, and clearly presents the need for more accessible sources of compatible soil data to facilitate broad-scale syntheses, as well as the challenges of providing such a resource. However, the description of the process of data harmonization and aggregation is somewhat confusing in the text (Fig. 3 provides an excellent visual summary). Please see the specific comments for more details. Additionally, the metadata template only provides sparse instructions for how to properly fill it out. I had to carefully compare a few of the source data files with the accompanying (filled out) metadata template in order to understand exactly what data was required for the metadata template and in what format it should be provided. A more thorough guide or an additional supporting document would improve this process. For example, the ISRaD database (Lawrence et al., 2020) mentioned in this manuscript provides both a template file and a supporting "template information file" to facilitate data entry. I would recommend implementing a solution along these lines.

One important feature of a synthesized database is data transparency, which SoDaH excels at in some ways, but falls short in others. The workflow for aggregating data preserves the raw data, which is the gold standard. However, with the way the metadata template is structured there is no clear way to document the data source for site level data, which data contributors enter manually. As multiple contributors could provide data from the same site, it seems possible that conflicting data could be reported for, say, mean annual temperature. How would users distinguish which reported value is more appropriate for their analysis?

Finally, a critical feature of SoDaH is the web-based tool for querying and generating reports from the database (Shiny app), but unfortunately the use of this tool is neither well documented in the text, nor by the supporting resources (with the exception of an hour-long webinar available as a downloadable video clip, which while very useful, is not very user-friendly). Underselling this extremely powerful feature of SoDaH is in my view the biggest shortcoming in SoDaH as presented here. Providing a simplified overview or vignette that gives an example of the kinds of queries that can be made

(filters, etc.) and the reports that can be generated (visualizations, downloadable .csv tables, maps, etc.) would greatly help with reception and use of SoDaH within the community.

With minor improvements to the clarity of the text, and some additional documentation of the usage of the Shiny app, I think the manuscript is an excellent candidate for publication in ESSD.

Specific comments

Lines 58-60: Terms such as "harmonize" and "automate" would benefit from explicit definitions, although understandably this may not be possible with the word limits of the abstract (perhaps in the main text?).

Line 83: It would be helpful to expand on or quantify exactly what you mean by "similar data products". Expanding on the importance of public availability of these databases would also be helpful.

Lines 123-130: Fine as is, but perhaps this information could be simplified in the text and expanded on in a table?

Line 145: While the framework for reporting data from experimental manipulations is a key asset of SoDaH, it is not clear why SoDaH allows for a greater range of spatial and temporal data than other databases.

Line 168: Suggest moving "Fig. 3" to the end of the sentence (after "...the structure of SoDaH")

Line 180: The use of "ontologies" in this context is not entirely unclear and sounds like jargon. Additionally, what does "automatic harvesting of data" mean? Do you mean something more along the lines of automating the process of data acquisition? I realize those are similar, but the meaning is not clear from how it is written.

Lines 182-185: It seems like the "site" (or "location"?) is the fundamental organizational

unit of SoDaH. It might be helpful to state that more clearly and expand on the example provided in order to help readers understand how to define a site/location and how that definition relates to data organization at each level of SoDaH, i.e. from raw data to querying the aggregated database.

Lines 186-189: This section is not clear to me. What do you mean by the statement that data on the location tab are "broadcast to every row of the harmonized dataset"? The analogy that clarified this somewhat for me (used elsewhere in the manuscript) is the idea that the profile tab is a "map" for matching variables in the raw data onto the standardized variables in SoDaH. Additionally it is not clear why or when it would be appropriate to move data from the site to the profile tab.

Line 194: Can you expand upon (either in the text or by providing an example in the supporting information) how one would go about describing additional aggregation steps and how that would be implemented in the data aggregation process? This seems like a very messy and case-by-case basis, but also like a problem that would be encountered fairly frequently.

Line 200: When and how (what platform) would users "point to the Google Drive directory"? I assume this means when running the function in R?

Line 201: Suggest "...generates a new flat file(s) in which the relevant variable names and units are standardized..."

Line 202-206: If possible, it would be helpful to define or clarify some of the terms you use throughout this section in advance, e.g. "harmonized dataset", "Level-1" data products". The process is very clearly shown in Fig. 3 along with the terminology, so perhaps you could give a one-sentence description of the workflow in which you name the outputs of each step of the process?

Lines 208-212: This may not be the best place for it, but some discussion of data transparency would help to showcase the strengths SoDaH. Aside from the issue of

site-level data lacking a clear source, the preservation of raw data in SoDaH is a valuable feature. However, it is not clear from the website how to access the raw data files (I was able to find them, but it wasn't simple). Perhaps this could be clarified or stated explicitly somewhere?

Line 239: Clarify in the template how to specify these grouping variables.

Lines 251-255: Perhaps histograms of these data could replace Fig. 5? In its current format Fig. 5 is completely illegible.

Table 1: Consider providing some examples of gradient studies or time series.

Figure 3: Excellent figure, very clearly describes the data workflow.

Figure 5: This figure is illegible and it is unclear what it shows. Suggest replacing with histograms of site characteristics (see comment for lines 251-255).

---

## Referee Comment (RC2) · Anonymous Referee #2 · 3 Dec 2020

I very much enjoyed reading about and exploring the new database: The Soils Data Harmonization (SoDaH) database. SoDaH is a valiant effort to combine the soil carbon data from three massive scientific efforts (LTER's, CZO's, and NEON) and to create a database structure that allows for time series and experimental data. Gradient data were also mentioned as something new to include though I do not see where gradients would have had trouble fitting into existing database structures like ISCN. The database uses a similar hierarchal structure to existing databases such as ISCN and ISRaD, and the familiarity should help the greater soil carbon community both contribute data and use the database.

Overall the description of the database in this manuscript was pretty clear in terms of how the database is structured (with the exception of layers). I did find, however, that more information is needed about the expectations of data contributors and users. I will go into more detail on that below. Lastly, I applaud the inclusion of the web-based shiny app. I enjoyed exploring the data with it, and I think it will help people easily see whether the data that they seek exists in the database and if it does, what the data coverage is. I recently spent a long time struggling to access and understand the data from a certain plant trait database, and I could see how the experience would have been much better with a shiny app. I will warn the authors that my comments go beyond the paper, to the webpage, shiny and git repository. With these ESSD papers that one has to evaluate the whole package.

Line Edits

Line 55: Get rid of comma on after "Synthesizing these data"

Line 135-140: The description of ISRaD makes is sound like 13C was a goal of ISRaD, though in reality ISRaD focuses on radiocarbon and includes 13C data if available, but datasets with only 13C data were not targeted. Furthermore, ISRaD includes 14CO2 data from gas wells, incubations, and fluxes. I think a more accurate description would be "radiocarbon from bulk soils, soil fractions, and soil gases."

Line 167: Is raw data the correct term here? To me raw data implies that the data is straight from an instrument and may still be in peak heights or areas and not corrected to actual carbon values. However, I am not sure what would be better to call it.

Line 171: I think what you mean by "layer" should be described here. It is also unclear how the layer fits in within the profile tab, or is it its own tab? It is hard to tell because it is a different color than profile in figure 2. I guess if there is no fraction data, then layer does not need to be its own tab but there did seem to be fraction data included based on the fields in the shiny app.

ESSDD
Line 182-189: More concrete examples might be helpful here as it seems to me that some studies will only have a single location to describe (an experiment) and then the treatments would be described in the profile tab, but a gradient study might have multiple location tabs or would the lat and long fields have to be moved to the profile tab in that case? I think the latter is described on line 189, but clarification would be good when it comes to gradients. For NEON data is every terrestrial site in its own google drive folder as single locations or are they all combined into one folder?

194: Can you define what you mean by "stacked". I am pretty sure it means that the from the same experiment the soil carbon and nitrogen data would each get its own line if they were on separate raw data files. This seems to be another case where a description of a concrete example would help.

197: It is unclear who the intended users of the soilHarmonization R package are. Is it the database managers or are the data contributors expected to use this package?

210: Why is the dataHarvest function not part of the above R package? Or is it? Again, is the data contributor expected to use this function after submitting data via their google drive folder? If they are not, who views the QC? Would it be best for the data contributor to view it since they know their data best?

225: I did not see many R scripts in this git repository, which seems to include the main paper. Is this the right address?

240: For users of this database, how can they access the grouping variable information? It does not seem like users can view these templates directly? Or maybe they can, and I just could not find that info?

279: Future contributions from who? Who will be overseeing this database? Is there a steering committee or manager? How will succession in such positions be handled?

280: It was hard to find how to contribute data on the website since it was towards the bottom of the database tab, maybe make it its own link at the top like Authorship is?

ESSDD
Also looking through the instructions it was not clear how to handle layer. Maybe it's just me, but a description of a study and an example of a filled-out template could be helpful here. I am really stuck on how layers should be described.

Figure 1: Can DIRT and nutnet also be touching the green circle because they are manipulations?

Figure 2: There are two locations shown here. Do they each get their own Location tabs?

Figure 5: Can the depth axis have units or at least put the units in the caption?

Other questions:

Where are these level 0 data stored? It seems like the contributions are given via users' own google drive folders, so that does not seem very permanent.

The authorship process is very clear on the website and seems to pertain to future users of the data, but the policy is not mentioned at all in this paper. Should it be?

For the Shiny app, I wanted more information on how to interpret each dataset's (level 1) QAQC. I looked at data I am familiar with and could not really understand what the graphs were trying to show.

Is there a way to only download the data that you query in the shiny app? Or could the shiny app show the code used for a certain query to help the user subset the downloaded database in R?

**ESSDD**

---

## Author Comment (AC1) · 24 Dec 2020

Comments from Jeffrey Beem Miller, Referee #1, are provided below in normal text. Our responses to each are below each comment in bold with suggested changes to the revised manuscript identified by text in quotes. Note, this is easier to see in the .pdf files we've uploaded along with this plain text response.

General comments

The authors present a new database (SoDaH) of soil observations synthesized from datasets curated by five well-known research networks: LTER, CZO, NEON, NutNet,

and DIRT. Two key aspects of SoDaH make it unique among the new generation of soil databases: 1) flexibility in its approach to harmonizing data from diverse sources, and 2) the development of a powerful web-based tool for querying, visualizing, and extracting subsets of data. These two features of SoDaH make it a valuable addition to the growing pool of soil databases available to the soil science community. Thank you, we appreciate recognition of these novel aspects of the database.

The manuscript is generally well written, and clearly presents the need for more accessible sources of compatible soil data to facilitate broad-scale syntheses, as well as the challenges of providing such a resource. However, the description of the process of data harmonization and aggregation is somewhat confusing in the text (Fig. 3 provides an excellent visual summary). Please see the specific comments for more details. Additionally, the metadata template only provides sparse instructions for how to properly fill it out. I had to carefully compare a few of the source data files with the accompanying (filled out) metadata template in order to understand exactly what data was required for the metadata template and in what format it should be provided. A more thorough guide or an additional supporting document would improve this process. For example, the ISRaD database (Lawrence et al., 2020) mentioned in this manuscript provides both a template file and a supporting "template information file" to facilitate data entry. I would recommend implementing a solution along these lines. This echoes comments made by R2. We have modified the website to include a new 'contribute data' tab (https://lter.github.io/som-website/contribute_data.html) that includes: Instructions provided here give an overview of how to contribute data. This link provides access to the SoDaH database template file Note that you will need to copy the database template file to be able to edit it. An example google directory, with primary data and a completed metadata template is provided for reference. One important feature of a synthesized database is data transparency, which SoDaH excels at in some ways, but falls short in others. The workflow for aggregating data preserves the raw data, which is the gold standard. However, with the way the metadata template is structured there is no clear way to document the data source for site level data, which data contribu-

tors enter manually. As multiple contributors could provide data from the same site, it seems possible that conflicting data could be reported for, say, mean annual temperature. How would users distinguish which reported value is more appropriate for their analysis? This comment is 100% accurate. Any synthesis is only as good as the data that is contributed to it. Manually entering data into the metadata template does introduce potential sources of error, but by creating a scripted infrastructure to generate the harmonized level-2 data we can go back to correct errors that may have been introduced with the level-0 contributions. As such, we hope users of the database will let the steering committee know when they find discrepancies or inconsistencies in the data.

Finally, a critical feature of SoDaH is the web-based tool for querying and generating reports from the database (Shiny app), but unfortunately the use of this tool is neither well documented in the text, nor by the supporting resources (with the exception of an hour-long webinar available as a downloadable video clip, which while very useful, is not very user-friendly). Underselling this extremely powerful feature of SoDaH is in my view the biggest shortcoming in SoDaH as presented here. Providing a simplified overview or vignette that gives an example of the kinds of queries that can be made (filters, etc.) and the reports that can be generated (visualizations, downloadable .csv tables, maps, etc.) would greatly help with reception and use of SoDaH within the community. Thank you for the care in looking at the work we've contributed as part of SoDaH. We agree that the shiny app is really powerful and suggest the following text to help guide users through its functionality

"To facilitate user interaction with the SoDaH database, and to provide a simplified approach for data queries and analysis, we developed a web-based application using R Shiny (Chang et al. 2020). This SoDaH application is publicly accessible and hosted by the National Center for Ecological Analysis and Synthesis (NCEAS) at https://cosima.nceas.ucsb.edu/lter-som (last accessed July 15, 2020; source code: https://github.com/lter/lterwg-som-shiny). With the SoDaH application, users can perform a number of tasks to aid data discovery, visualization and analysis. We provide a brief description of this resource that highlights key features of the R Shiny SoDaH application. In the Query section of the application, the top portion of the page provides a variety of data filter options to assist users with partitioning the database. Specifically, users may subset the database by any combination of research network, experiment type, and soil depth, while also specifying whether they wish to include or exclude experimental treatments or time-series data. Below the filter options, the Output section of the page contains three separate features arranged into labeled application tabs. The Plot tab allows users to quickly create basic analysis plots (point, histogram, or boxplot) using both covariates (e.g., Fe concentration) and metadata (e.g., mean annual precipitation). In the Map tab, users may specify which analyte in the database to display on a spatial map. Numeric values are symbolized using a color gradient and the interactive map functionality allows users to both adjust the map scale and select from numerous basemap options. Finally, the Table tab provides users with the ability to directly view, search and download the user-specified data subset as a flat file (.csv). The plot, map and table features are all responsive to user specified changes in the data filters and will update in realtime. In the Data Summary section of the SoDaH application, two feature tabs are provided to help users identify the data available for a specific site or analyte. The By Analytes tab allows users to view the number of analyte values that exist across all of the unique sites in the database. Users may specify up to four different analytes at a time to be included in the summary table output. The By Site tab allows users to view all of the analyte data available for a specific site. As the amount of data may be quite large for some sites, options are provided to narrow the summary output to include only profile, location or character class data. The SoDaH application also includes a Data Key section, where users may view a full copy of the metadata template used for the SoDaH database construction, including descriptions of database fields and their associated metadata. The searchable key is split into two sections, location and profile, in the same manner as the metadata template used to describe raw data for the harmonization process. Field names in the provided key

match exactly with analyte and metadata options provided in the Plot and Map features in the Query section of the application. Finally, the application provides a Comments section where users may submit an inquiry about the database or the application."

With minor improvements to the clarity of the text, and some additional documentation of the usage of the Shiny app, I think the manuscript is an excellent candidate for publication in ESSD. Thank you for this positive assessment

Specific comments

Lines 58-60: Terms such as "harmonize" and "automate" would benefit from explicit definitions, although understandably this may not be possible with the word limits of the abstract (perhaps in the main text?). We removed these phrases from the abstract. In the main text 'automated' is replaced with 'scripted', which is more accurate. We also clarify that "In the harmonized dataset, we convert analyte names and units to a standard output"

Line 83: It would be helpful to expand on or quantify exactly what you mean by "similar data products". Expanding on the importance of public availability of these databases would also be helpful. We now define these similar data syntheses. "Providing similar data syntheses with information on soil carbon and associated covariates (e.g., climate, productivity, and soil physical and chemical properties) in public databases is critical to advancing understanding soil biogeochemistry."

Lines 123-130: Fine as is, but perhaps this information could be simplified in the text and expanded on in a table? Given the data we're actually collecting are well documented on the website, metadata template, and Shiny App we'll avoid provided in table here that may give readers an abbreviated understanding of the data SoDaH contains.

Line 145: While the framework for reporting data from experimental manipulations is a key asset of SoDaH, it is not clear why SoDaH allows for a greater range of spatial and temporal data than other databases. Reviewer #2 raised similar concerns. We've

now clarified that "data from these kinds of studies should be incorporated into existing database structures, like ISCN, but the additional metadata requested as part of SoDaH helps database users understand more information about how data were collected from individual studies."

Line 168: Suggest moving "Fig. 3" to the end of the sentence (after "...the structure of SoDaH") Done

Line 180: The use of "ontologies" in this context is not entirely unclear and sounds like jargon. Additionally, what does "automatic harvesting of data" mean? Do you mean something more along the lines of automating the process of data acquisition? I realize those are similar, but the meaning is not clear from how it is written. We suggest replacing 'automated' with 'scripted' here and clarify "Ultimately, sophisticated metadata, such as controlled vocabularies and other, more expressive semantic technologies, may facilitate scripted harvesting of data from disparate networks and repositories (e.g., see review by Buck et al. 2019 for trends and examples in Marine Science).

Lines 182-185: It seems like the "site" (or "location"?) is the fundamental organizational unit of SoDaH. It might be helpful to state that more clearly and expand on the example provided in order to help readers understand how to define a site/location and how that definition relates to data organization at each level of SoDaH, i.e. from raw data to querying the aggregated database. Reviewer #2 raised similar questions: We suggest expanding this paragraph as follows: The metadata template in SoDaH matches site-level information with the detailed measurements collected at each study site. Data on the location tab represents site characteristics for a single site or location (e.g., Prospect Hill Warming experiment at Harvard Forest). Accordingly, the harmonization script broadcasts data provided on the location tab (latitude, longitude, mean annual temperature, etc.) to every row of the harmonized dataset. Data on the profile tab includes profile information about experimental levels (e.g., plots within experimental blocks) and experimental treatments (e.g. +N fertilization) that help clarify how the data were collected. Data on the profile tab should also correspond to columns of variables that are reported in the Level-0 data (e.g., soil organic C measured at different soil layers). Accordingly, the harmonization script copies each unique measurement from the profile tab into a column of data in the harmonized dataset. Data contributors, therefore, can move variables from the location to profile tabs when appropriate. For example, NutNet and NEON data were submitted to SoDaH with information from multiple sites on a single .csv file that provided information about each site as unique columns of data. We, therefore, moved site information (e.g., climate, latitude and longitude) onto the profile tab for these networks. Similarly, gradient studies that report tabular data for individual soil profiles can move information on slope, aspect, vegetation communities or parent material (typically on the location tab) onto the profile tab of the metadata template.

Lines 186-189: This section is not clear to me. What do you mean by the statement that data on the location tab are "broadcast to every row of the harmonized dataset"? The analogy that clarified this somewhat for me (used elsewhere in the manuscript) is the idea that the profile tab is a "map" for matching variables in the raw data onto the standardized variables in SoDaH. Additionally it is not clear why or when it would be appropriate to move data from the site to the profile tab. Please see response to the previous comment, above.

Line 194: Can you expand upon (either in the text or by providing an example in the supporting information) how one would go about describing additional aggregation steps and how that would be implemented in the data aggregation process? This seems like a very messy and case-by-case basis, but also like a problem that would be encountered fairly frequently. We're still working on a robust way to do this, but will code to the github repository. However, because SoDaH is a flat database values from these different data files will be stacked, meaning that information from different Level-0 datasets would be recorded in different rows of the aggregated Level-2 database (in the example above, soil properties and productivity will be included, but in different rows). Additional aggregation steps, therefore, may be required to align data within

sites. Users can find this information in the database column labeled merge_align, which is a logical that identifies if multiple data files can be merged. Notes under columns align_1 and align_2 are intended to help communicate what common data fields can help with this alignment (e.g. experimental or treatment levels, L1 and tx_L1, respectively). To help users understand the database column information, the complete database key is provided in the SoDaH online application and gives users descriptions of the column contents.

We also note in section 3.3: "As mentioned in section 2.4, future contributions of code to analyse the SoDaH database are encouraged. These contributions should be made to the LTER SOM GitHub repository, with a priority on developing additional utilities to align and aggregate datasets from individual sites and locations. Contributions will be reviewed by the SoDaH steering committee (currently Wieder, Pierson and Earl) and made publicly available. The committee will continue oversight while new funding options and/or partnerships (e.g., ISCN) are explored.

Line 200: When and how (what platform) would users "point to the Google Drive directory"? I assume this means when running the function in R? This is clarified in section 2.3 "We developed the soilHarmonization package in R (R Core Team 2020) to harmonize and aggregate the SoDaH database"

Line 201: Suggest "...generates a new flat file(s) in which the relevant variable names and units are standardized..." Done

Line 202-206: If possible, it would be helpful to define or clarify some of the terms you use throughout this section in advance, e.g. "harmonized dataset", "Level-1" data products". The process is very clearly shown in Fig. 3 along with the terminology, so perhaps you could give a one-sentence description of the workflow in which you name the outputs of each step of the process? This may be appropriate as the last paragraph of section 2.1. The workflow for synthesizing is summarized in Figure 3 and in the following sections. Briefly, Primary data (Level-0) are identified by data

providers and variables are mapped to standardized units and vocabulary using the metadata templates (section 2.2). These data are harmonized into Level-1 data with soil harmonization script that renames variables, conducts unit conversions, and performs quality control checks (section 2.3). Finally, Level-1 data are aggregated into the Level-2 dataset, which can be visualized with the SoDah R Shiny app and queried with data analysis tools (section 2.4).

Lines 208-212: This may not be the best place for it, but some discussion of data transparency would help to showcase the strengths SoDaH. Aside from the issue of C4 site-level data lacking a clear source, the preservation of raw data in SoDaH is a valuable feature. However, it is not clear from the website how to access the raw data files (I was able to find them, but it wasn't simple). Perhaps this could be clarified or stated explicitly somewhere? In section 2.2 we note: "These primary data may or may not be in a published state but, if not published, would be equivalent to data provided for publication. Many of the datasets in SoDaH were already published in public repositories like EDI, the repository for LTER data, or available through the NEON data portal. Users can find these primary data using the doi provided for the individual dataset in the harmonized dataset. Other datasets that we wanted to include in SoDaH, however, had not been published or were difficult to find or identify (mainly data from CZO sites and the DIRT network, but also some LTER data). Publishing these primary data remains an active priority for our working group."

From a reproducibility standpoint, we probably should be storing the completed metatemplates in Zenodo, or add them to the current database repository in EDI. We wonder, however, how much value would be gained from such an effort?

Line 239: Clarify in the template how to specify these grouping variables. In section 2.2 we note "Additional aggregation steps, therefore, may be required to align data within sites. Users can find this information in the database column labeled merge_align, which is a logical that identifies if multiple data files can be merged. Notes under columns align_1 and align_2 are intended to help communicate what common data

fields can help with this alignment (e.g. experimental or treatment levels, L1 and tx_L1, respectively)." and reference this section here in the text.

Lines 251-255: Perhaps histograms of these data could replace Fig. 5? In its current format Fig. 5 is completely illegible. "Our intent with this figure is to illustrate the number of sites in each network, the temporal length of their data record, and the depth to which soils are typically sampled" With respect, we'd prefer the figure as-is to illustrate these points and clarify the intent in the figure caption

Table 1: Consider providing some examples of gradient studies or time series. This seems to make the most sense in the table heading "Gradient studies may include measurements along a hillslope catena (e.g., several CZO sites), across vegetation communities (typically LTER sites), or surveys intended to capture local- to regional-variability (especially NEON periodic soil sampling). Time series studies involve re-peated measurements in the same sites over time (LTER and NEON) and they which may also include experimental manipulations (e.g., NutNet, DIRT, & LTER)."

Figure 3: Excellent figure, very clearly describes the data workflow. Thank you

Figure 5: This figure is illegible and it is unclear what it shows. Suggest replacing with histograms of site characteristics (see comment for lines 251-255). See also our response, above.

Please also note the supplement to this comment:
https://essd.copernicus.org/preprints/essd-2020-195/essd-2020-195-AC1-supplement.pdf

---

## Author Comment (AC2) · 24 Dec 2020

Comments from Anonymous Referee #2 are provided below in normal text. Our responses to each are below each comment in bold with suggested changes to the revised manuscript identified by text in quotes. Note, this is easier to see in the .pdf files we've uploaded as a summlement along with this plain text response.

I very much enjoyed reading about and exploring the new database: The Soils Data Harmonization (SoDaH) database. SoDaH is a valiant effort to combine the soil carbon data from three massive scientific efforts (LTER's, CZO's, and NEON) and to create a database structure that allows for time series and experimental data. Gradient data

were also mentioned as something new to include though I do not see where gradients would have had trouble fitting into existing database structures like ISCN. The database uses a similar hierarchal structure to existing databases such as ISCN and ISRaD, and the familiarity should help the greater soil carbon community both contribute data and use the database. Thanks for this supportive comment. We note that Reviewer #1 raised similar questions, which we clarify: "Data from these kinds of studies (including gradient studies) should be incorporated into existing database structures, like ISCN, but the additional metadata requested as part of SoDaH helps database users understand more information about how data were collected from individual studies."

Overall the description of the database in this manuscript was pretty clear in terms of how the database is structured (with the exception of layers). I did find, however, that more information is needed about the expectations of data contributors and users. I will go into more detail on that below. Lastly, I applaud the inclusion of the web-based shiny app. I enjoyed exploring the data with it, and I think it will help people easily see whether the data that they seek exists in the database and if it does, what the data coverage is. I recently spent a long time struggling to access and understand the data from a certain plant trait database, and I could see how the experience would have been much better with a shiny app. I will warn the authors that my comments go beyond the paper, to the webpage, shiny and git repository. With these ESSD papers that one has to evaluate the whole package. We are happy that you explored and appreciated the "unpublished" features of SoDaH that we have created to facilitate use of and, hopefully, contributions to the database. As much as possible, we have taken the suggestions provided, which are summarized below.

Line Edits Line 55: Get rid of comma on after "Synthesizing these data" Done

Line 135-140: The description of ISRaD makes is sound like 13C was a goal of ISRaD, though in reality ISRaD focuses on radiocarbon and includes 13C data if available, but datasets with only 13C data were not targeted. Furthermore, ISRaD includes 14CO2 data from gas wells, incubations, and fluxes. I think a more accurate description would

be "radiocarbon from bulk soils, soil fractions, and soil gases." We've changed the description of ISRaD to include "radiocarbon from bulk soils, soil fractions, and soil gases".

Line 167: Is raw data the correct term here? To me raw data implies that the data is straight from an instrument and may still be in peak heights or areas and not corrected to actual carbon values. However, I am not sure what would be better to call it. This is tricky, and now define "These primary data may or may not be in a published state but, if not published, would be equivalent to data provided for publication"

Line 171: I think what you mean by "layer" should be described here. It is also unclear how the layer fits in within the profile tab, or is it its own tab? It is hard to tell because it is a different color than profile in figure 2. I guess if there is no fraction data, then layer does not need to be its own tab but there did seem to be fraction data included based on the fields in the shiny app. This is illustrated in Figure 2. These are good questions we seek to clarify with the following revised text.

"To simplify the workflow for data contributors, the metadata template only includes a single tab each for location and profile data. Within these tabs, data contributors are able to add information on metadata (found on the 'location' tab) and layer or fraction data (found on the 'profile' tab; Fig. 2). Layer data includes information on soil chemical and physical properties that may be measured on bulk soils for defined soil horizons or depth increments. Fraction data would include similar measurements on defined fractions within individual soil layers (e.g. percent soil organic carbon on density fractionated soils). Note, SoDaH currently has sparse data from measured soil fractions, which have therefore been omitted from Fig 2 for simplicity, but the database structure can include information on soil fractions.

Line 182-189: More concrete examples might be helpful here as it seems to me that some studies will only have a single location to describe (an experiment) and then the treatments would be described in the profile tab, but a gradient study might have multiple location tabs or would the lat and long fields have to be moved to the profile tab in that case? I think the latter is described on line 189, but clarification would be good when it comes to gradients. For NEON data is every terrestrial site in its own google drive folder as single locations or are they all combined into one folder? Reviewer 1 raised similar concerns. We agree, more examples would help clarify this text: "The metadata template matches site-level information with the detailed measurements collected at each study site. Data on the location tab represents site characteristics for a single site or location (e.g., Prospect Hill Warming experiment at Harvard Forest). Accordingly, the harmonization script broadcasts data provided on the location tab (latitude, longitude, mean annual temperature, etc.) to every row of the harmonized dataset. Data on the profile tab includes profile information about experimental levels (e.g. plots within experimental blocks) and experimental treatments (e.g. +N fertilization) that help clarify how the data were collected. Data on the profile tab should also correspond to columns of variables that are reported in the Level-0 data (e.g., soil organic C measured at different soil layers). Accordingly, the harmonization script copies each unique measurement from the profile tab into a column of data in the harmonized dataset. Data contributors, therefore, can move variables from the location to profile tabs when appropriate. For example, NutNet and NEON data were submitted to SoDaH with information from multiple sites on a single .csv file that provided information about each site as unique columns of data. We, therefore, moved site information (e.g., climate, latitude and longitude) onto the profile tab for these networks. Similarly, gradient studies that report tabular data for individual soil profiles can move information on slope, aspect, vegetation communities or parent material (typically on the location tab) onto the profile tab of the metadata template."

194: Can you define what you mean by "stacked". I am pretty sure it means that the from the same experiment the soil carbon and nitrogen data would each get its own line if they were on separate raw data files. This seems to be another case where a description of a concrete example would help. Another good suggestion we seek to clarify: "However, because SoDaH is a flat database values from these different data

files will be stacked, meaning that information from different Level-0 datasets would be recorded in different rows of the aggregated Level-2 database (in the example above, soil properties and productivity will be included, but in different rows). Additional aggregation steps, therefore, may be required to align data within sites. This can be accomplished with information from experimental levels and experimental treatments."

197: It is unclear who the intended users of the soilHarmonization R package are. Is it the database managers or are the data contributors expected to use this package? Either would be appropriate. "The package includes functions that harmonize Level-0 data into Level-1 data. Data contributors or database managers use the data_harmonizaiton function tools to read and harmonize user-provided raw data that are mapped to a metadata template with controlled vocabulary and standard units (Fig. 3)."

210: Why is the dataHarvest function not part of the above R package? Or is it? Again, is the data contributor expected to use this function after submitting data via their google drive folder? This function is not part of the package above, as "This function is intended for use by database managers". But the repository with this function is provided.

If they are not, who views the QC? Would it be best for the data contributor to view it since they know their data best? This comments to the harmonization package (previous paragraph), which we clarify "These Level-1 data products are stored in the same Google Drive directory as the Level-0 data with resulting output identified with a modified filename. This allows data contributors and database managers to verify the QC report and ensure appropriate data harmonization."

225: I did not see many R scripts in this git repository, which seems to include the main paper. Is this the right address? More scripts are available in the main repository, https://github.com/lter/lterwg-som, but the link provided in the manuscript is intended to include more curated examples, especially from published papers. Since this is the

first SoDaH manuscript we don't have much to share yet.

240: For users of this database, how can they access the grouping variable information? It does not seem like users can view these templates directly? Or maybe they can, and I just could not find that info? "Users can find this information in the database column labeled merge_align, which is a logical that identifies if multiple data files can be merged. Notes under columns align_1 and align_2 are intended to help communicate what common data fields can help with this alignment (e.g. experimental or treatment levels, L1 and tx_L1, respectively). To help users understand the database column information, the complete database key is provided in the SoDaH online application and gives users descriptions of the column contents."

279: Future contributions from who? Who will be overseeing this database? Is there a steering committee or manager? How will succession in such positions be handled? As mentioned in section 2.4, future contributions of code to analyse the SoDaH database are encouraged. These contributions should be made to the LTER SOM GitHub repository, with a priority on developing additional utilities to align and aggregate datasets from individual sites and locations. Contributions will be reviewed by the SoDaH steering committee (currently Wieder, Pierson and Earl) and made publicly available. The committee will continue oversight while new funding options and/or partnerships (e.g., ISCN) are explored

280: It was hard to find how to contribute data on the website since it was towards the bottom of the database tab, maybe make it its own link at the top like Authorship is? Also looking through the instructions it was not clear how to handle layer. Maybe it's just me, but a description of a study and an example of a filled-out template could be helpful here. I am really stuck on how layers should be described. This is a good idea, also suggested by Reviewer #1. We've updated the website, as requested (https://lter.github.io/som-website/contribute_data.html) and included more information (see text above) clarifying what is included in Layer data.

Figure 1: Can DIRT and nutnet also be touching the green circle because they are manipulations? This is a good suggestion we can modify in revisions

Figure 2: There are two locations shown here. Do they each get their own Location tabs? Yes, we've clarified this in the caption: "The right side of the figure illustrates data from two hypothetical locations (e.g., a LTER and CZO site, respectively) where Location 1 includes data from two profiles that each have information from one layer. Location 2 provides data from one profile that has information from three layers. Any location may provide data from multiple profiles or layers. With data harmonization data for each profile and layer will inherit metadata and location data that are provided in the location tab." Figure 5: Can the depth axis have units or at least put the units in the caption? This has been done

Other questions: Where are these level 0 data stored? It seems like the contributions are given via users' own google drive folders, so that does not seem very permanent. Yes, a copy of the primary data for harmonization are referenced in a google drive folder, which is not a permanent repository. That's why we note in 2.2: "These primary data may or may not be in a published state but, if not published, would be equivalent to data provided for publication. Many of the datasets in SoDaH were already published in public repositories like EDI, the repository for LTER data, or available through the NEON data portal. Other datasets that we wanted to include in SoDaH, however, had not been published or were difficult to find or identify (mainly data from CZO sites and the DIRT network, but also some LTER data). Publishing these primary data remains an active priority for our working group."

From a reproducibility standpoint, we probably should be storing the completed metatemplates in Zenodo, or add them to the current database repository in EDI. We wonder, however, how much value would be gained from such an effort?

And in section 3.4: "We ask that new contributions of primary data that are harmonized into SoDaH be published with a unique DOI".

The authorship process is very clear on the website and seems to pertain to future users of the data, but the policy is not mentioned at all in this paper. Should it be? The authorship policy was mainly for our working group as we developed SoDaH. Now that the dataset is published "We encourage users of SoDaH data to cite both this publication and the dataset citation provided by the EDI data portal in their products"

For the Shiny app, I wanted more information on how to interpret each dataset's (level 1) QAQC. I looked at data I am familiar with and could not really understand what the graphs were trying to show. The "data summary" tab has data " by site" that includes 'notes.pdf' information. These were used by the database managers to check the data harmonization process. Now that the database is published, this information isn't really needed and we have removed these links from the Shiny app.

Is there a way to only download the data that you query in the shiny app? Or could the shiny app show the code used for a certain query to help the user subset the downloaded database in R?

Yes, "the data table on the Query page of the SoDaH Shiny application is responsive to the filter options at the top of the Query page. When users click the "Download data" button next to the table, the downloaded .csv file will contain the same data shown in the application table at that time. Code examples for working with the database, including how to filter by specific column values, are provided in the GitHub repository (https://github.com/lter/lterwg-som/data-processing/Tarball_v2 scripts)."

Please also note the supplement to this comment:
https://essd.copernicus.org/preprints/essd-2020-195/essd-2020-195-AC2-supplement.pdf

——————————————————

---

## Author Response (AR2)

Figs 1 and 3 reflect changes requested by reviewer #2.